# Outcome of High-Dose Chemotherapy Followed by Autologous Stem Cell Transplantation in Relapsed/Refractory Hodgkin Lymphoma after Different Numbers of Salvage Regimens

**DOI:** 10.3390/cells13020118

**Published:** 2024-01-09

**Authors:** Jacopo Mariotti, Francesca Ricci, Laura Giordano, Daniela Taurino, Barbara Sarina, Chiara De Philippis, Daniele Mannina, Carmelo Carlo-Stella, Stefania Bramanti, Armando Santoro

**Affiliations:** 1Department of Oncology and Hematology, IRCCS Humanitas Research Hospital, Via Manzoni 56, 20089 Milan, Italy; francesca.ricci@humanitas.it (F.R.); daniela.taurino@humanitas.it (D.T.); barbara.sarina@humanitas.it (B.S.); chiara.de_philippis@humanitas.it (C.D.P.); daniele.mannina@humanitas.it (D.M.); carmelo.carlostella@hunimed.eu (C.C.-S.); stefania.bramanti@humanitas.it (S.B.); armando.santoro@humanitas.it (A.S.); 2Department of Biomedical Sciences, Humanitas University, Via Rita Levi Montalcini 4, 20072 Milan, Italy; 3Biostatistic Unit, IRCCS Humanitas Research Hospital, Via Manzoni 56, 20089 Milan, Italy; laura.giordano@humanitas.it

**Keywords:** Hodgkin lymphoma, transplantation, new drugs

## Abstract

The introduction of novel drugs (*PD-1* inhibitors and/or brentuximab vedotin) into salvage regimens has improved the response rate and the outcome of patients with relapsed/refractory Hodgkin lymphoma. However, the impact of new drugs on the outcome has not been adequately investigated so far. We retrospectively analyzed 42 consecutive patients treated at our institution with high-dose chemotherapy/autologous stem cell transplantation after either one standard chemotherapy represented by BEGEV (*n* = 28) or >1 salvage therapy (ST) comprising novel drugs (*n* = 14). With a median follow-up of 24 months, the 2-year cumulative incidence of relapse was similar between the two cohorts: 26% for 1 ST and 18% for >1 ST (*p* = 0.822). Consistently, overall survival and progression-free survival did not differ among the two groups: 3-year overall survival was 91% and 89% (*p* = 0.731), respectively, and 3-year progression-free survival was 74% and 83% (*p* = 0.822) for only one and more than one salvage regimens, respectively. Of note, the post-transplant side effects and engraftment rates were similar between the 1 ST and >1 ST cohorts. In conclusion, consolidation with high-dose chemotherapy/autologous stem cell transplantation is a safe and curative option, even for patients achieving disease response after more than one rescue line of therapy.

## 1. Introduction

High-dose chemotherapy (HDT) followed by autologous stem cell transplantation (ASCT) represents the standard of care treatment for relapsed/refractory (R/R) Hodgkin lymphoma (HL) [1]. This recommendation is based on the results of two randomized clinical trials showing improved outcomes for patients undergoing salvage chemotherapy followed by consolidation with HDT/ASCT relative to conventional chemotherapy alone [2,3]. Nevertheless, only 50% of patients with refractory HL can be cured by HDT/ASCT [1,4], and the outcome may vary based on several prognostic factors, such as primary refractory disease [5], relapse within 12 months of initial treatment, extranodal disease, bulky disease, active disease at the time of transplant and the presence of B symptoms [6,7].

The outcome of patients with R/R HL has improved in recent years due to the introduction of novel target therapies and a new combination of chemotherapies. First-line salvage treatment with BEGEV (bendamustine, gemcitabine, vinorelbine) resulted in a 5-year progression-free survival (PFS) of 77% [8]. Brentuximab vedotin (BV) as the first salvage treatment was associated with an overall response rate (ORR) > 80% [9,10,11]; however, the complete remission (CR) rate was <40%, and 5-year overall survival (OS) and PFS were 41% and 22%, respectively [12]. Consistently, *PD-1* inhibitors provided excellent ORR [12], regardless of previous exposure to BV [13], but CR rates were particularly low (16–17%) [13,14]. The combination of BV and *PD-1* inhibitors as the first salvage treatment was very promising, yielding a 3-year PFS of 77% for patients undergoing ASCT [15], a result consistent with the results achieved with BEGEV. In accordance with all these findings, the Stanford University has recently showed an improved outcome for patients with R/R HL undergoing HDT/ASCT between 2011 and 2020 relative to the previous decade, due to better salvage therapies either before or after ASCT [16].

While consolidation with HDT/ASCT is a well-established approach in cases of response to first salvage therapy, the best strategy for patients refractory to the front-line salvage regimen for R/R HL is still a matter of debate [17]. In patients with chemo-refractory disease, many centers have employed novel target therapies as a second-line salvage regimen, followed by consolidation with allogeneic stem cell transplantation (Allo-SCT) when disease response was achieved. This strategy succeeded at the expense of increased potential toxicity, especially when checkpoint inhibitors (CPIs) [18] were employed. More recently, promising outcomes have been described for patients receiving HDT/ASCT as a consolidation regimen after salvage therapies with BV [17,19] or CPI [20,21]. Nevertheless, it is not well known whether patients undergoing consolidation with HDT/ASCT beyond first-line salvage therapy (ST) have similar outcomes to those receiving only one salvage line. This question is relevant to draw the best strategy for this subset of R/R HL patients.

In order to address this question, we conducted a retrospective analysis at our institution, comparing the outcome of patients undergoing HDT/ASCT after either first-line salvage chemotherapy or two or more lines of therapy comprising new target therapies.

## 2. Patients and Methods

### 2.1. Setting and Design

This is a retrospective, monocentric study conducted at the Humanitas Cancer Center (Rozzano, Italy), a referral center for oncological disease treatment in northern Italy. Consecutive patients with R/R HL and treated with HDT and ASCT from January 2018 to December 2022 were identified according to the following inclusion criteria: patients with primary refractory or relapsed HL receiving ASCT as consolidation treatment; patients achieving either complete or partial remission (PR) after a salvage regimen; and patients receiving either second-line chemotherapy followed by HDT/ASCT or more than one salvage regimen followed by HDT/ASCT. Exclusion criteria comprised patients receiving HDT/ASCT not in CR or PR; patients being treated with CPI or BV as front-line therapy for HL; patients receiving tandem ASCT-allogeneic transplantation; and those receiving Allo-SCT as a consolidation regimen. In total, 42 patients were analyzed. Of them, 28 (67%) received one line of salvage treatment, while 14 (33%) received two or more lines. Of note, 11 patients were excluded from the analysis because their pre-transplant disease status was represented either by stable (SD, *n* = 6) or progressive disease (PD, *n* = 5).

The Institutional Review Board of the Humanitas Cancer Institute approved the study. The study code is ONC/OSS-23/2023. Approval date was 12 December 2023; approval code: 47/2023. All procedures were performed in line with the Helsinki Declaration. All patients signed informed consent for the use of their data for research purposes.

### 2.2. HL Diagnosis and Definitions

HL was diagnosed by performing a biopsy of a lymph node or extranodal disease when present. In the case of relapsed or refractory disease, a new biopsy was usually performed, especially when new disease sites occurred. All patients had a diagnosis of classical HL according to previously reported criteria [22]. In particular, diagnosis was based on the identification of neoplastic cells that were further immunophenotypically characterized by using the following markers: CD30, CD15, CD3, CD20, MUM1, PAX5, ALK and EBV/LMP1. Refractory HL was defined according to previous publications [23] by progression at any time during chemotherapy and up to 3 months after the end of chemotherapy, by failure to achieve at least PR with first-line therapy or by the persistence of significant (score 4 or 5/5) residual FDG metabolic activity using the quantitative 5-point scale Deauville score.

### 2.3. Salvage Therapy, Autologous Stem Cell Harvest and HDT

All patients within the cohort undergoing HDT and ASCT after one ST received BEGEV chemotherapy (bendamustine 90 mg/mq on days 1–2, gemcitabine 800 mg/mq on days 1 and 4, vinorelbine 20 mg/mq on day 1) [8]. Among the patients receiving HDT and ASCT after two or more STs, most received BEGEV, while one was treated with IGEV chemotherapy (ifosfamide 2 g/m^2^ on days 1–4, gemcitabine 800 mg/mq on days 1 and 4, vinorelbine 20 mg/mq on days 1 and 4). Non-chemotherapy drugs comprised BV 1.8 mg/kg every 3 weeks, pembrolizumab 200 mg every 3 weeks or a combination treatment consisting of BV plus pembrolizumab or BV plus bendamustine. Stem cell harvest was attempted in all patients during the first salvage chemotherapy by administering G-CSF at a dosage of 5 µg/kg starting on day +7 after the beginning of chemotherapy. The minimum number of stem cells required to proceed to HDT was 2 × 10^6^ CD34/kg. HDT consisted of FEAM (fotemustine 150 mg/mq on days 1–2, etoposide 200 mg/mq on days 3–6, cytarabine 400 mg/mq on days 3–6 and melphalan 140 mg/mq on day 7) in all patients.

All patients received prophylaxis against bacterial, virus and fungal infections. Antimicrobial prophylaxis was begun at the hospital during the conditioning regimen and consisted of acyclovir at 800 mg per day and levofloxacin at 500 mg per day. Low-dose oral fluconazole (100 mg/day) was started during the conditioning regimen and withdrawn at the end of the aplastic phase.

### 2.4. Engraftment

Neutrophil engraftment was defined as the first of three consecutive days with an absolute neutrophil count (ANC) of 0.5 × 10^9^/L after transplantation. Platelet engraftment was defined as a platelet count of 20 × 10^9^/L, with no transfusions during the preceding 7 days.

### 2.5. Routine Surveillance and Infection Definitions

All patients were placed in HEPA-filtered single rooms from the beginning of the conditioning until neutrophil recovery (defined as ANC > 0.5/L). They received a low microbial diet. For blood vessel access, a central venous line was placed. In the case of fever (defined when the temperature was >38 °C), the following examinations were performed: duplicate blood cultures; a high-resolution thorax CT scan; and bronco-alveolar lavage when possible, in the case of lung abnormality. Pneumonia was diagnosed when a new lung area of consolidation or ground glass was identified via a CT scan according to the forementioned schedule. Bacteriemia was diagnosed when a bloodstream infection was identified via blood cultures performed as previously described. Septic shock was defined according to Quick SOFA criteria when vasopressors were required to maintain a median arterial pressure ≧ 65 mmHg plus a serum lactate level >2 mmol/L in the absence of ipovolemia.

### 2.6. Statistical Analysis

Categorical variables are expressed as numbers and proportions, and continuous variables are expressed as medians with the respective range. OS was calculated as the time from transplant to death from any cause or to the last follow-up for patients alive. PFS was calculated as the time from transplant to relapse/progression, death by any cause or last follow-up. OS and PFS were estimated and are displayed using the Kaplan–Meier method [24]. Non-relapse mortality (NRM) was defined as death from transplant-related side effects. For the calculation of NRM, disease relapse or progression was treated as a competing event. In contrast, NRM was the competing event for calculating the cumulative incidence (CI) of relapse or progression [25]. Univariate analyses were performed, the log-rank test was used to compare the groups in terms of PFS and OS, and Grey’s test was used to compare cumulative incidence. Statistical significance was set at a level of 0.050. A statistical analysis was performed with SAS v. 9.4 and R v. 4.2.

## 3. Results

### 3.1. Patients’ Characteristics

The patients’ characteristics are summarized in Table 1. The median age of the whole population was 35 years: the patients receiving one chemotherapy only (1 ST) were slightly older than the ones in the >1 ST cohort (37 vs. 32 years old). Male/female gender was equally represented in the two groups. Lugano stage and the presence of extranodal disease at diagnosis were similar between the two cohorts. The Lugano stage and extranodal site consistently did not differ at the time of disease relapse or progression. All patients had responsive disease and were either in CR (*n* = 41; 98%) or PR (*n* = 1; 2%). The two cohorts were comparable in terms of refractory versus relapsed disease and time to relapse (≤12 vs. >12 months; *p* = 0.294).

In total, 14 patients received non-chemotherapy drugs. Among them, nine received two lines, four received three lines, and one patient had four lines of therapies before consolidation. The salvage lines are summarized in Figure 1.

### 3.2. Engraftment

The median time to achieve unsupported ANC counts was 15 days for both the cohort receiving 1 ST (range 12–54) and the one treated with >1 ST (range 12–27; *p* = 0.442). Consistently, the 30-day cumulative incidence of engraftment did not differ among the two cohorts (93% vs. 100%; *p* = 0.523), as reported in Figure 2A. Of note, the median number of infused CD34+ stem cells was similar among the two cohorts: 5.5 × 10^6^/kg (range: 3.0–9.6) vs. 5.6 × 10^6^/kg (range: 2.3–7.6; *p* = 0.857).

The median time to achieve platelet counts above 20,000/µL was 14 days in both cohorts (*p* = 0.216), and the 30-day cumulative incidence of platelet engraftment was 85% vs. 100% for the 1 ST cohort vs. the >1 ST group (Figure 2B, *p* = 0.093).

### 3.3. ASCT Toxicity

Toxicity after HDT was quite similar in the two cohorts. The most common adverse events were oral mucositis, which was grade 2 (G2) or 3 (G3) in most patients, and diarrhea, which was usually G2 or G3 (Table 2). Infectious complications consisted of fever of unknown origin in most cases with no difference between the 1 ST line vs. >1 ST cohorts (Table 2). Other infectious complications comprised pneumonia, bacteriemia and septic shock. Of note, there were more viral reactivations in the cohort of subjects treated with only one line of chemotherapy, but this difference was not statistically significant (*p* = 0.863).

### 3.4. Outcome after HDT/ASCT

The median follow-up in the whole population was 24 months, with no statistically significant difference between the two groups (24 vs. 23 months, *p* = 0.512). Five patients had relapsed in the 1 ST cohort compared with two patients in the >1 ST group at a median time of 10 months (range: 5.4–27.7). The cumulative incidence of relapse was comparable among the two cohorts: the 2-year cumulative incidence was 26% versus 18% (*p* = 0.823; Figure 3A) for the 1 ST cohort vs. >1 ST group, respectively.

Two patients died, one from each cohort, due to adverse events of Allo-SCT, which was performed for disease relapse.

Three-year OS and PFS for the whole population were 91% and 77%, respectively. OS was similar in the two groups: 3-year OS was 91% for the single-salvage-chemotherapy cohort and 89% for the >1 ST group (*p* = 0.731; Figure 3B). Consistently, PFS did not differ between the two cohorts: 3-year PFS was 74% in the one-chemotherapy-salvage group and 83% in the >1 ST cohort (*p* = 0.822; Figure 3C).

No patients experienced NRM in the two cohorts.

The univariate analysis did not show any association between the main demographic variables and the outcome for both cohorts (Table 3).

## 4. Discussion

Even with the limits of its design and small sample size, the results of this retrospective analysis provide further evidence for the curative potential of HDT/ASCT consolidation treatment for R/R HL when disease response is achieved beyond the first salvage line with novel target therapies. Our analysis adds to the existing literature by directly comparing the outcomes of two cohorts of patients undergoing HDT/ASCT after either a standard first salvage line comprising a chemotherapy combination or multiple rescue lines comprising BV and/or CPI.

The introduction of BV and CPI has revolutionized the outcome of patients with R/R HL. HDT/ASCT has been traditionally reserved for patients with chemosensitive relapse due to a more favorable outcome for those achieving a complete metabolic response at the time of PET scan [26]. In the pre-novel agent’s era, only a small fraction of patients refractory to first-line ST were rescued by strategies employing tandem autologous [27] or tandem autologous–allogeneic transplantation [28,29]. Since the introduction of novel agents as a second- or third-salvage line of therapy, the best consolidation strategy has been a matter of debate. Some centers prefer BV or CPI treatment until progression, but 5-year follow-ups of BV and CPI found that only 9% of patients receiving BV monotherapy [12] and 20% of those treated with CPI [30] achieve long-term remission. Other centers have successfully performed consolidation with Allo-SCT given the chemo-refractory nature of the disease, either after BV [31] or CPI [18] rescue, achieving a 3-year PFS of at least 40% [32]. A higher risk of NRM hampers the success of Allo-SCT compared with BV and CPI therapy or with ASCT. Recent studies have suggested that CPI may have a chemosensitizing effect [33,34], with an ORR of 66–82% achieved with a rechallenge with IGEV or BEGEV chemotherapy, allowing for R/R HL patients to be bridged to Allo-SCT. Given the renovated chemosensitivity, HDT/ASCT was rechallenged as a consolidation strategy in R/R HL rescued with CPI, showing a promising 18-month PFS and OS of 81% and 96%, respectively [21].

The results reported in our retrospective report may extend these data because they support the long-term efficacy of HDT/ASCT consolidation for patients achieving disease response with novel agents and suggest that this strategy yields results similar to those of conventional first-line ST comprising chemotherapy combinations. Of note, our 3-year PFS of 77% and 2-year cumulative incidence of relapse of 18% compare well with the results obtained by Merryman et al. [21] and Casadei et al. [20] in the setting of patients undergoing ASCT after CPI salvage therapy, who found a PFS of 81% and 73% at 18 months and 5 years, respectively.

The results of our study are also consistent with those of two other recent reports comprising CPI or BV in the armamentarium of salvage therapies before ASCT. In one, Spinner et al. [16] performed a large (*n* = 342 patients) single-center-based retrospective analysis comparing the outcome of R/R HL patients undergoing HDT/ASCT between either 2001 and 2010 or 2011 and 2020. This analysis allowed the authors to compare the results achieved with chemotherapy-only salvage strategies or with novel agents comprising BV (57% of patients) or CPI (31% of the patients). Similar to our study, patients undergoing HDT/ASCT in more recent decades had received more lines of salvage therapy than those treated in past decades where chemotherapy represented the only rescue strategy. PFS and OS were similar for patients receiving HD/ASCT after achieving CR either in the older or in the modern era. The outcomes observed in our small cohort compare well with those reported by Spinner et al. [16]: for the cohort receiving novel agents, 3-year OS and PFS were 89% and 83% in our study vs. the 4-year OS and PFS of 89% and 73% in the Stanford analysis. Spinner et al. [16] also reported an enhanced outcome for the cohort of patients receiving pre-ASCT immunotherapy relative to those receiving platinum-based chemotherapy: 4-year PFS was 91% vs. 66%. This observation is similar to that reported by a multicenter study comprising 854 patients with R/R undergoing HDT/ASCT across the United States [35]. In this report, Desai et al. found that patients receiving the combination of BV/nivolumab had higher CR rates and PFS relative to those receiving conventional chemotherapy, while those receiving only BV had lower CR rates. Due to the small size of our cohort, we were not able to compare the outcomes of patients receiving BV or CPI before ASCT: only two patients relapsed in the >1 ST group, one after BV and one after CPI.

Our study has several limitations, namely, the retrospective nature of the analysis and the small sample size. Due to the lack of randomization, the two cohorts are essentially composed of different patients. Nevertheless, the presence of similar pre-transplant characteristics allowed for a statistical comparison between the two groups. Unfortunately, we were not able to increase the size of our cohort in order to achieve more conclusive results either to compare standard chemotherapy vs. novel agents or to compare the outcomes of patients receiving BV vs. CPI. Moreover, our cohort was too small to identify adverse characteristics discouraging HDT/ASCT as a consolidation strategy. Merryman et al. [21] found a less favorable outcome (PFS = 51%) for subjects not responding to *PD-1* inhibitors. In our study, the only patient rechallenged with BEGEV chemotherapy after a *PD-1* inhibitor failed, relapsed 3 months after HDT/ASCT, even if transplanted in CR. Another limitation of our study is represented by the follow-up. Even though a 24-month median follow-up may capture most events in terms of HL relapse, we may have missed late events due to toxicity or disease relapse, as recently underlined by the publication of the long-term follow-up results of the EORTC/LYSA/FIL H10 trial for localized HL [36].

Salvage therapy with CPIs before Allo-SCT was described to be associated with an increased risk of fatal hyper-acute graft-versus-host disease (GVHD), non-infectious febrile episodes requiring corticosteroids and other immune-related adverse events [37,38]. No data have been published so far in the setting of ASCT. Our results show that rescue treatment with novel agents before HDT/ASCT is not associated with an increased risk of adverse events.

In conclusion, HDT/ASCT retains its curative potential for R/R HL when disease response is achieved beyond the first line. Consolidation with HDT/ASCT should be prioritized over Allo-SCT, given the reduced incidence of adverse events and mortality rate. Whether patients not responding to novel target therapies and achieving CR with chemotherapy rechallenge are a better candidate for consolidation with HDT/ASCT or Allo-SCT needs to be investigated in larger cohorts of clinical trials.

## 5. Conclusions

The results of this study add to the growing literature supporting consolidation with HDT/ASCT when disease response, either CR or PR, is achieved in R/R HL, even after >1 line of salvage therapy. Our data support the consideration of new strategies combining novel drugs, particularly CPI, with chemotherapy in the second-line setting, particularly for patients with primary refractory disease.

## Figures and Tables

**Figure 1 cells-13-00118-f001:**
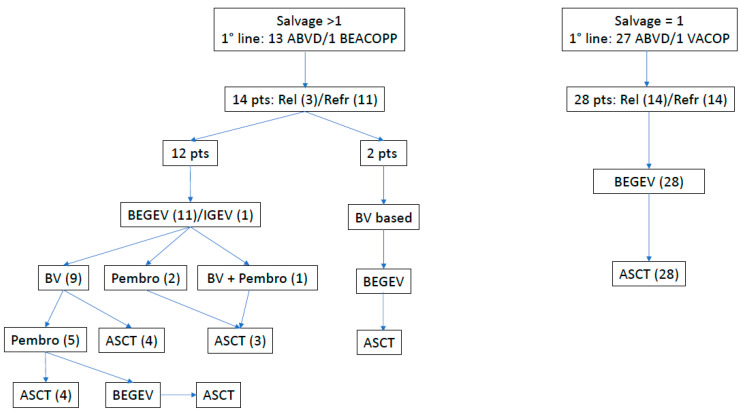
Treatment schedule before HDT/ASCT. Detailed sequence of salvage treatments performed in patients with Rel or Ref Hodgkin lymphoma. The one-salvage-line cohort received only chemotherapy consisting of BEGEV or IGEV. The >1 salvage line group received, besides chemotherapy, new target therapy drugs (BV or Pembro) as rescue before autologous stem cell transplantation (ASCT). ABVD: adriamycin, bleomycin, vinblastine, dacarbazine; ASCT: autologous stem cell transplantation; BEACOPP: doxorubicin, cyclophosphamide, etoposide, procarbazine, prednisolone, vincristine, bleomycin; BEGEV: bendamustin, gemcitabine, vinblastine; BV: brentuximab vedotin; IGEV: ifosfamide, etoposide, vinblastine; VACOP: doxorubicin, cyclophosphamide, etoposide, prednisolone, vincristine, bleomycin; Pembro: pembrolizumab; Pts: Patients; Ref: refractory; Rel: relapsed.

**Figure 2 cells-13-00118-f002:**
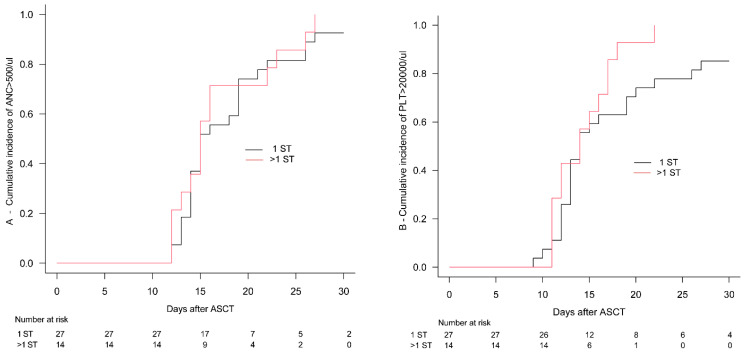
Cumulative incidence of engraftment. The 30-day cumulative incidence of neutrophil (**A**) and platelet (**B**) engraftment in the 1 salvage chemotherapy cohort (1 ST: black line) and in >1 rescue treatment (comprising brentuximab vedotin and checkpoint inhibitors) group (>1 ST: red line). ANC: absolute neutrophil count; ASCT: autologous stem cell transplantation; PLT: platelet; ST: salvage therapy.

**Figure 3 cells-13-00118-f003:**
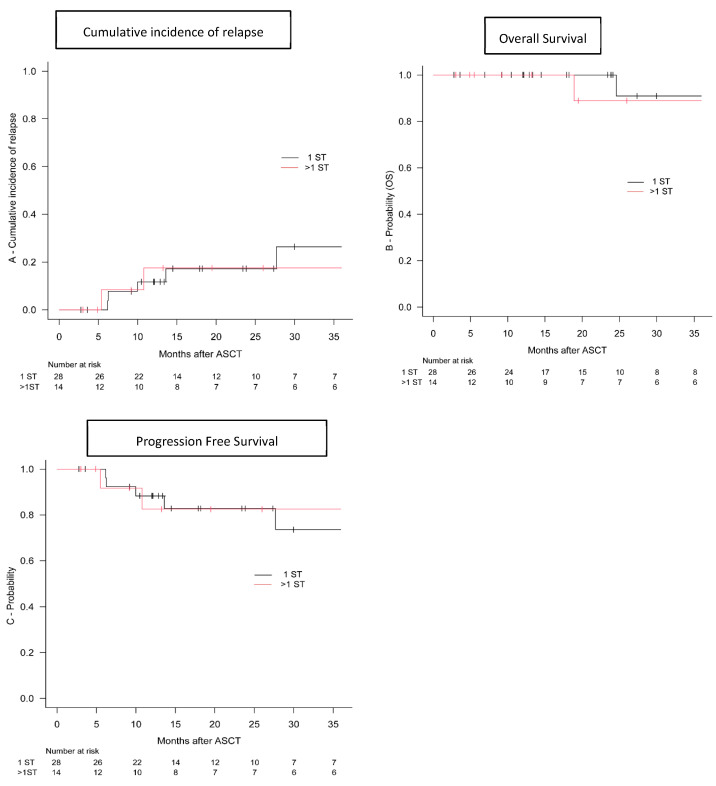
Outcome of Hodgkin lymphoma patients with relapsed or refectory disease. The 2-year cumulative incidence of relapse (**A**) after ASCT in the 1 salvage line (1 ST: black line) vs. >1 rescue treatment cohorts (>1 ST: red line); Kaplan–Meier estimate of 3-year OS (**B**) and progression-free survival (**C**) in the 1 salvage line (1 ST: black line) vs. >1 rescue treatment cohorts (>1 ST: red line). ASCT: autologous stem cell transplantation; OS: overall survival; ST: salvage therapy.

**Table 1 cells-13-00118-t001:** Patients’ characteristics.

	All, *n* (%)	1 Salvage (BEGEV Only) *n* (%)	>1 Salvage (New Drugs) *n* (%)	*p*-Value
N patients	42 (100)	28(100)	14 (100)	-
Follow-up (months), median (range)	24 (3–58)	24 (3–56)	23 (3–58)	0.512
Age (years), median (range)	35 (18–71)	37 (20–64)	32 (18–71)	0.163
Sex, F/M	21/21	14/14	7/7	1.000
Stage at diagnosis:				0.100
● I–II	21 (51%)	11 (39%)	10 (71%)	
● III–IV	20 (49%)	16 (61%)	4 (29%)	
Extranodal site at diagnosis:				0.520
● No	23 (56%)	14 (52%)	9 (64%)	
● Yes	18 (44%)	13 (48%)	5 (36%)	
Disease status pre-auto-transplantation:				0.333
● CR	41 (98%)	28 (100%)	13 (93%)	
● PR	1 (2%)	0	1 (7%)	
Relapsed/refractory:				0.294
● Refractory	25 (60%)	14 (50%)	11 (79%)	
● Relapsed ≤ 12 months	11 (26%)	9 (32%)	2 (14%)	
● Relapsed > 12 months	6 (14%)	5 (18%)	1 (7%)	
Stage at relapse/refractory:				0.277
● I–II	30 (71%)	18 (64%)	12 (86%)	
● III–IV	12 (29%)	10 (36%)	2 (14%)	
Extranodal site at relapse/refractory:				1.000
● No	32 (76%)	21 (75%)	11 (79%)	
● Yes	10 (24%)	7 (25%)	3 (21%)	
*n* alvage lines before auto-transplantation:				<0.001
● 1	28 (67%)	28 (100%)	-	
● 2	9 (21%)	-	9 (64%)	
● 3	4 (10%)	-	4 (29%)	
● 4	1 (2%)	-	1 (7%)	

Table legend: BEGEV: bendamustine, gemcitabine, vinorelbine; CR: complete remission; F: female; M: male; PR: partial remission.

**Table 2 cells-13-00118-t002:** Salvage therapy cohort.

	1 Salvage (BEGEV Only) *n* (%)	>1 Salvage (New Drugs) *n* (%)	*p*-Value
Mucositis:			0.526
● Grade 1	5 (18%)	4 (29%)	
● Grade 2	3 (11%)	3 (21%)	
● Grade 3	19 (68%)	7 (50%)	
● None	1 (3%)	0	
Diarrhea:			0.963
● Grade 1	5 (18%)	2(15%)	
● Grade 2	5 (18%)	3 (21%)	
● Grade 3	3 (10%)	1 (7%)	
● None	15 (54%)	8 (57%)	
Infections:			0.939
● Fever of unknown origin	19 (54%)	11 (58%)	
● Pneumonia	5 (14%)	2 (14%)	
● Bacteremia	4 (10%)	3 (21%)	
● Septic shock	1 (4%)	1 (7%)	
● Virus reactivation	4 (10%)	1 (7%)	
● Campylobacter enteritis	1 (4%)	1 (7%)	
● Other severe infections	1 (4%)	–	

Table legend: BEGEV: bendamustine, gemcitabine, vinorelbine.

**Table 3 cells-13-00118-t003:** Univariate analysis of main variables affecting relapsed/refractory Hodgkin lymphoma patients’ outcomes.

	3-Year OS (%)	*p*-Value	3-Year PFS (%)	*p*-Value	2-Year CIR (%)	*p*-Value
Stage at diagnosis:		0.746		0.526		0.522
● I–II	94		79		21	
● III–IV	88		75		25	
ExN at diagnosis:		0.731		0.715		0.714
● No	91		84		16	
● Yes	89		72		28	
Stage at relapse/refractory:		0.569		0.216		0.223
● I–II	89		84		16	
● III–IV	100		40		60	
ExN at relapse/refractory:		0.459		0.484		0.473
● No	88		78		22	
● Yes	100		75		25	
Type of salvage:		0.731		0.822		0.822
● 1 salvage (BEGEV)	91		74		26	
● >1 salvage (new drugs)	89		83		18	
Relapse/Refractory:		0.800		0.635		0.627
● Refractory	100		83		17	
● <12 months	83		88		13	
● >12 months	93		72		29	

Table legend: BEGEV: bendamustine, gemcitabine, vinorelbine; CIR: cumulative incidence of relapse; ExN: extranodal sites; OR: overall survival; PFS: progression-free survival.

## Data Availability

The data presented in this study are available on the Zenodo platform, doi: 10.5281/zenodo.10153251.

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
