# Peer review of "Outcome of High-Dose Chemotherapy Followed by Autologous Stem Cell Transplantation in Relapsed/Refractory Hodgkin Lymphoma after Different Numbers of Salvage Regimens"

_cells, 2024, doi:10.3390/cells13020118_

Round 1

Reviewer 1 Report

Comments and Suggestions for Authors

The manuscript is well-written and the results are convincing.  The limitation of this study is small sample size.  According to "Patients and Methods", patients receiving HDT/ASCT not in CR or PR were excluded from this cohort.  Did these patients receive salvage therapy? Do you have any data on OS and PFS for this group of patients?   

Author Response

The manuscript is well-written and the results are convincing.  The limitation of this study is small sample size.  According to "Patients and Methods", patients receiving HDT/ASCT not in CR or PR were excluded from this cohort.  Did these patients receive salvage therapy? Do you have any data on OS and PFS for this group of patients?

We thank the reviewer for these comments. There were 11 patients that received HDT/ASCT either with stable (SD) or progressive disease (PD) status. Of these, 5 were in PD and 6 in SD. Three patients received standard salvage chemotherapy followed by HDT/ASCT while they were in SD. All of them relapsed and died because of HL. The other 8 patients were included in tandem Auto-Allo program: 5 of these patients achieved Allo-SCT after brentuximab vedotin or nivolumab salvage therapy. Five patients are actually alive in complete remission, while 3 had died: 1 of relapse and 2 of post-transplant complications.

We mentioned the patients excluded from the analysis in the Methods section, lines 86-88

Reviewer 2 Report

Comments and Suggestions for Authors

The authors present analysis of outcomes after ASCT at a large referral center, focusing analysis on the outcomes after greater than one salvage regimen with the incorporation of brentuximab vedotin and/or PD-1 inhibitor treatment.  The results are of some interest supporting ASCT in this setting (>1 line of treatment with novel agents with a CR) and are consistent with other recent publications. 

1. A large study recently showed improvement in outcomes with ASCT in the era of novel agents PMID : 36857637.  This should be referenced in the introduction and/or discussion

2. For Figure 3 the labeling should be more clear, particularly for panel C, to help the reader distinguish what data is being depicted in the survival curve.

3. Overall the impact is limited by the heterogeneity of the cohort and small size, but the findings still have merit and the analysis is soundly conducted

Author Response

The authors present analysis of outcomes after ASCT at a large referral center, focusing analysis on the outcomes after greater than one salvage regimen with the incorporation of brentuximab vedotin and/or PD-1 inhibitor treatment.  The results are of some interest supporting ASCT in this setting (>1 line of treatment with novel agents with a CR) and are consistent with other recent publications. 

  1. A large study recently showed improvement in outcomes with ASCT in the era of novel agents PMID : 36857637.  This should be referenced in the introduction and/or discussion

We thank the reviewer for this comment, we agree that it is mandatory to mention the results of this article. We quoted the article in the introduction (lines 52-55) and in the discussion (lines 291-312) sections.

  1. For Figure 3 the labeling should be more clear, particularly for panel C, to help the reader distinguish what data is being depicted in the survival curve.

We added the highlights for each panel of Figure 3 in order to make it more clear as suggested.

  1. Overall the impact is limited by the heterogeneity of the cohort and small size, but the findings still have merit and the analysis is soundly conducted.

We agreed with the reviewer, and we thank for the appreciation of our findings even with the limits of the small and heterogeneous cohort. We further underline the limits of our study in the discussion (lines 313-320)

Reviewer 3 Report

Comments and Suggestions for Authors

This articles is very interesting and relevant in the domain of Oncology specially the Hodgkin Lymphoma. It was well designed and executes as well. However, it must addressed the following comments before consideration for publication.

1.      Author may mention the technique used to diagnose the Hodgkin Lymphoma and Relapsed/Refractory Hodgkin Lymphoma.

2.      The finding would be more conclusive with more sample size i.e.  Number of patient.  

3.      Author also need to mentions the technique used to diagnose pneumonia, bacteremia and septic shock.

4.      Rearrange the tables. Currently it looks like all merged in one.

thanks.

Manish

Author Response

This article is very interesting and relevant in the domain of Oncology specially the Hodgkin Lymphoma. It was well designed and executes as well. However, it must addressed the following comments before consideration for publication.

  1. Author may mention the technique used to diagnose the Hodgkin Lymphoma and Relapsed/Refractory Hodgkin Lymphoma.

We thank the reviewer for this fundamental comment. We detailed technique and criteria used in the Methods section, lines 94-105.

  1. The finding would be more conclusive with more sample size e.  Number of patient. 

We agree with the reviewer, but unfortunately we were not able to extend the sample size. We further underline the limits of our study in the discussion (lines 313-320)

  1. Author also need to mention the technique used to diagnose pneumonia, bacteremia and septic shock.

We thank the reviewer for this comment. We added a section in Methods to detail antimicrobial prophylaxis and diagnostic criteria, lines 132-143.

  1. Rearrange the tables. Currently it looks like all merged in one.

      We rearranged the Tables as correctly pointed out.